# Epidemiological study of the relationship between meteorological factors and onset of acute aortic dissection in Japan

**Ayami Ishikawa[1,2], Yasuto Sato[1]\*, Yasuhiko Terai[3], Takeshi Usui[1]**

**1** Graduate School of Public Health, Shizuoka Graduate University of Public Health, Shizuoka, Japan,
**2** Department of Nursing, Shizuoka City Shizuoka Hospital, Shizuoka, Japan, **3** Department of
Cardiovascular Surgery, Shizuoka City Shizuoka Hospital, Shizuoka, Japan

\* yssato@s-sph.ac.jp

## Abstract

Several factors influence the onset of acute aortic dissection (AAD). However, few studies
have examined AAD onset, weather conditions, and meteorological factors in Japan. This
study aimed to identify meteorological factors associated with the onset of AAD in Japan. In
this self-controlled study, patients diagnosed with AAD onset from May 1, 2012, to April 30,
2021, at Shizuoka City Shizuoka Hospital (Shizuoka, Japan) were included. Meteorological
data from the Shizuoka District Meteorological Office were used. Control days were ran-
domly selected from a 29-day period centered on the day of onset. Conditional logistic
regression models were used to obtain odds ratios (ORs) and 95% confidence intervals
(CIs) for the onset of AAD relative to the control day due to changes in meteorological fac-
tors. In total, 538 patients were included. The meteorological factors associated with the
onset of AAD were identified as the daily mean temperature (OR = 1.10; 95% CI = 1.04–
1.16), daily minimum temperature (OR = 1.09; 95% CI = 1.03–1.14), daily maximum temper-
ature (OR = 1.05; 95% CI = 1.00–1.10), and the mean of the daily mean temperatures for
the previous 7 days (OR = 1.17; 95% CI = 1.07–1.28) with a 1°C decrease in the tempera-
ture. The results of this study are expected to help raise awareness in clinical practice and
among the general public about the increased risk of AAD associated with a drop in
temperature.

## Introduction

Acute aortic dissection (AAD) is characterized by the aortic wall's dissection at the media
level, resulting in a longitudinal tear along the aorta and creating a two-chambered state [1].
The incidence of AAD is low, with 4.8 cases per 100,000 people annually [2]. However, it is a
serious disease with a reported fatality rate of 15% [3]. Given these characteristics, identifying
factors associated with the onset of AAD holds significant importance. These include internal
risk factors such as sex, smoking, hypertension, aortic diameter, and genetic factors, as well as
external risk factors such as physical stress and meteorological factors [1].

Epidemiological study of the relationship between
meteorological factors and onset of acute aortic
dissection in Japan. PLoS ONE 19(10): e0311489.

Universitaria Careggi, ITALY

**Data Availability Statement:** Clinical data can be
downloaded here. https://doi.org/10.5061/dryad.
12jm63z76 Meteorological data can be
downloaded from the Japan Meteorological

Agency website. https://www.data.jma.go.jp/risk/obsdl/index.php

**Funding:** The author(s) received no specific funding for this work.

**Competing interests:** The authors have declared that no competing interests exist.

Previous studies have investigated meteorological factors influencing the onset of AAD, including season [4–7], week [4], temperature [4, 6, 8], humidity [7], atmospheric pressure [4, 8], and wind speed [4]. Temperature has been reported as a meteorological factor affecting AAD onset, with the highest incidence in winter [4, 9]. Sudden cold exposure, such as that observed in winter, induces an increase in blood pressure, which, in turn, triggers the onset of AAD [10]. The difference between the highest and lowest daily temperatures also influences the onset of the disease [7]. However, some reports state that the effect of temperature on AAD onset is unclear [9]. The incidence of AAD was highest on Wednesdays and lowest on Sundays within the week [4]. Further, it has been reported that atmospheric pressure increases the incidence of AAD [4, 8]; however, no conclusions have been drawn, and the influence of humidity and wind speed on the development of AAD remains unknown [4].

Weather conditions and combinations of meteorological factors vary by country and region. Therefore, it is necessary to consider weather conditions in each region and clarify the meteorological factors that influence the onset of AAD. In Japan, the seasons in which AAD is most likely to occur are autumn and winter, and it has been reported that the meteorological factors that influence the onset of AAD are temperature and atmospheric pressure [11]. However, few studies have been conducted on the onset of AAD, weather conditions, and meteorological factors in Japan. This study aimed to identify the meteorological factors associated with the onset of AAD in Japan.

## Methods

### Study design

Previous studies have compared meteorological factors on days when AAD onset was observed and days when it was not [8, 12–14]. Approximately 7% of patients experiencing cardiopulmonary arrest upon hospital admission have Stanford A-type AAD [15]. This implies that on days without observed AAD onset, some patients may not be diagnosed with AAD due to death. Accordingly, this study employs a self-controlled study design comparing patients' data of the same time period with and without disease onset. Self-controlled study designs can be used when the exposure is temporary, and the onset of the outcome is sudden [16]. Personal factors of the individual patient that do not change over time, such as sex and genetic factors, naturally remain the same in both periods. Patient factors that change over time, such as age and history of smoking, were considered equivalent in this study because the comparison was within a short period of time.

The topography of Japan is long from north to south, and the country belongs to various climatic zones ranging from subarctic to subtropical. Furthermore, the weather conditions differ between the Pacific Ocean side and the Sea of Japan side. Shizuoka Prefecture, the subject of this study, is located in the Chubu region, the geographic center of Japan. Shizuoka Prefecture has a warm maritime climate due to its location on the Pacific Ocean [17]. The region experiences four distinct seasons with hot and highly humid summers and dry winters, characterized by the absence of significant temperature drops. The weather follows a cyclical pattern in spring and fall due to a migratory high-pressure system. The mean temperatures in Shizuoka Prefecture are highest in late July and lowest in late January. The daily mean temperature in Shizuoka Prefecture in late July 2020 was 28.2˚C, and in late January 2021 was 6.0˚C, a difference of 22.2˚C. This indicates that the daily mean temperature changes by approximately 0.8˚C per week.

Control days were randomly selected from a 29-day period centered on the AAD onset date to exclude the effects of annual changes in daily mean temperature. If the selection period of

the control date was widened, the seasons would be significantly different, so the selection period was set to within approximately 1 month.

## Data collection

Eligible patients were selected from the electronic medical records of Shizuoka City Shizuoka Hospital in Shizuoka City, Shizuoka Prefecture, over 9 years from May 1, 2012, to April 30, 2021. Patients were selected based on the diagnosis of "aortic dissection" and ICD-10 code (I71.0). As a result, 641 patients were selected. Patients with an AAD diagnosis that could not be confirmed in electronic medical records (n = 30), those for whom information on the date of onset was not available (n = 16), those with traumatic aortic dissection (n = 2), those admitted to Shizuoka Hospital after diagnosis and treatment at another hospital (n = 3), those with onset outside the target area (n = 6), those for whom information was not available (n = 29), patients with onset in the hospital (n = 7), and patients with onset outside the study period (n = 10) were excluded, leaving a total of 538 patients for analysis.

Patient data were obtained from electronic medical records and included sex, age at AAD onset, date and time of AAD onset, Stanford classification, complications, medication taken internally, medical history, family history of aortic disease, and history of smoking. The time of AAD onset was determined based on the "patient-reported symptom onset time" as documented in the initial medical record. For complications, data were collected on cardiopulmonary arrest, aortic rupture, cardiac tamponade, and malperfusion. Cardiopulmonary arrest was considered present if it was noted at the time of transport to the hospital. Aortic rupture, cardiac tamponade, and malperfusion were considered present if they were noted in the medical record at the time of admission. For malperfusion, only the presence or absence was reported, and the location could not be specified.

Meteorological data were obtained from the Shizuoka District Meteorological Office, which is the closest to Shizuoka City Shizuoka Hospital and is publicly available on the Internet for 3,652 days from January 1, 2012, to December 31, 2021, including the study period. The following data were obtained for this period: daily mean relative humidity, daily mean wind speed, sunshine duration, daily mean station pressure, daily mean temperature, daily minimum temperature, and daily maximum temperature. The definitions of the meteorological factors are as follows: The daily mean relative humidity is the mean humidity at the observation station every hour, 24 times a day. The daily mean wind speed is the mean of 144 daily values measured every 10 minutes. Sunshine duration is the sum of hourly observations of sunshine. The daily mean station pressure is the mean of the atmospheric pressure at the observation station every hour, 24 times a day. The daily mean temperature is the mean of the temperatures at the observation stations every hour, 24 times a day. The daily minimum temperature is the lowest of 144 daily values measured every 10 minutes. The daily maximum temperature is the highest of 144 daily values measured every 10 minutes.

## Statistical analysis

First, new variables were created from the collected data on meteorological factors: the difference between the daily maximum and minimum temperatures (diurnal temperature range), the difference in the daily mean temperature from the previous day, and the mean of the daily mean temperatures for the previous 3, 5, and 7 days (including the date of onset).

In the analysis, we first tabulated sex, age at AAD onset, season of AAD onset, time of AAD onset, Stanford Classification, complications, medications taken internally, medical history, family history of aortic disease, and history of smoking to show the distribution. Next, we examined whether there were any associations between metrological factors and AAD onset.

We calculated the mean, standard deviation, and p-value for case days and control days. Continuation conditional logistic regression models were used for each meteorological factor to obtain the odds ratios (ORs) and 95% confidence intervals (CIs) for the day of AAD onset (case days) relative to the control date (control days).

In the subset analyses, we examined the association between the mean of the daily mean temperatures for the previous 7 days and AAD onset. Subsets were created for men, women, high temperature period (June–September), low temperature period (December–March), under 65 years old, 65–74 years old, over 75 years old, type A dissection, type B dissection, history of smoking, complications of hypertension, family history of aortic disease, time of AAD onset of 0–6 am, time of AAD onset of 6 am–12 pm, time of AAD onset of 12–6 pm, time of AAD onset of 6 pm–12 am, men with a history of smoking, men with hypertension, and aged 65–74 years with hypertension. The significance level was set at 5%. R version 4.2.2 was used for the analysis.

**Ethical considerations.** This study was conducted in accordance with the Declaration of Helsinki. Further, this study was approved by the Ethics Committee for Medical Research of Shizuoka City Shizuoka Hospital (22–88), and permission to conduct the research was obtained from Shizuoka Graduate University of Public Health (22–72). The ethical committee approved the use of an opt-out consent process for study participation. An opt-out document was posted on the hospital's website, and individuals who did not agree to participate were excluded from the study. Data collection from medical records was conducted between November 1, 2022 and September 30, 2023. During data collection, it was possible to access patients' personal information from medical records. After data collection, analysis was performed without patients' personal information.

## Results

Table 1 shows the background information of the 538 eligible patients. There were more men (n = 307, 57.1%) than women (n = 231, 42.9%). The most frequent ages at AAD onset were 70–79 (n = 149, 27.7%) and 80–89 (n = 142, 26.4%). The most frequent season for AAD onset was winter (December–February), affecting 180 patients (33.5%). In terms of medical history, 158 patients (29.4%) had hypertension and 53 (9.9%) had aortic aneurysm. Twenty-three (4.3%) had a family history of aortic disease, and 197 (36.6%) had a history of smoking.

Table 2 shows the association between meteorological factors and AAD onset. Regarding the daily mean temperature (OR = 1.10; 95% CI = 1.04–1.16), daily minimum temperature (OR = 1.09; 95% CI = 1.03–1.14), and daily maximum temperature (OR = 1.05; 95% CI = 1.00–1.10), the temperature on the day of AAD onset was lower than that on the control day, and the OR was significantly increased. In terms of the mean of the daily mean temperatures for the previous 3, 5, and 7 days, the ORs for the previous 3 days (OR = 1.10; 95% CI = 1.03–1.18), 5 days (OR = 1.14; 95% CI = 1.05–1.24), and 7 days (OR = 1.17; 95% CI = 1.07–1.28) were significantly higher, and the OR tended to increase as the number of days increased.

Table 3 shows the association between the mean of the daily mean temperatures for the previous 7 days and AAD onset by subset analysis. The OR was significantly increased for men (OR = 1.25; 95% CI = 1.11–1.42), low temperature period (December–March) (OR = 1.27; 95% CI = 1.10–1.46], 65–74 years old [OR = 1.29; 95% CI = 1.08–1.54], type B dissection (OR = 1.33; 95% CI = 1.14–1.54), history of smoking (OR = 1.31; 95% CI = 1.12–1.53), complications of hypertension (OR = 1.25; 95% CI = 1.06–1.48), and time of AAD onset of 6 am–12 pm (OR = 1.24; 95% CI = 1.04–1.47]. In addition, in the combined subset analysis, the OR was significantly increased for "men with a history of smoking" (OR = 1.35; 95% CI = 1.15–1.59),

**Table 1. Characteristics of patients with acute aortic dissection (n = 538).**

| | | n (%) |
|---|---|---|
| Sex | Men | 307 (57.1%) |
| | Women | 231 (42.9%) |
| Age at AAD onset | 49 years or younger | 49 (9.1%) |
| | 50s | 63 (11.7%) |
| | 60s | 135 (25.1%) |
| | 70s | 149 (27.7%) |
| | 80s | 142 (26.4%) |
| Season of AAD onset | Spring (March–May) | 131 (24.3%) |
| | Summer (June–August) | 100 (18.6%) |
| | Autumn (September–November) | 127 (23.6%) |
| | Winter (December–February) | 180 (33.5%) |
| Time of AAD onset | 0–6 am | 67 (12.5%) |
| | 6 am–12 pm | 135 (25.1%) |
| | 12 pm–6 pm | 118 (21.9%) |
| | 6 pm–12 am | 92 (17.1%) |
| | No information | 126 (23.4%) |
| Stanford Classification | Type A | 334 (62.1%) |
| | Type B | 195 (36.2%) |
| | No information | 9 (1.2%) |
| Complications | No complications | 88 (16.4%) |
| (multiple answers) | Cardiac arrest, rupture, cardiac tamponade | 98 (18.2%) |
| | Malperfusion | 54 (10.0%) |
| | No information | 325 (60.4%) |
| Medications taken internally | No internal medications | 138 (25.7%) |
| (multiple answers) | Drugs for hypertension | 189 (35.1%) |
| | Drugs for dyslipidemia | 77 (14.3%) |
| | Fluoroquinolone antibacterial agents | 7 (1.3%) |
| | Steroids | 7 (1.3%) |
| | No information | 121 (22.5%) |
| Medical history | Hypertension | 158 (29.4%) |
| (multiple answers) | Aortic aneurysm | 53 (9.9%) |
| | Coronary artery disease | 50 (9.3%) |
| | Dyslipidemia | 42 (7.8%) |
| | Cerebrovascular disease | 42 (7.8%) |
| | Diabetes mellitus | 41 (7.6%) |
| | Chronic obstructive pulmonary disease | 16 (3.0%) |
| | Aortic dissection | 26 (4.8%) |
| | Bicuspid aortic valve | 3 (0.6%) |
| | Connective tissue disease | 3 (0.6%) |
| | No information | 26 (4.8%) |
| Family history of aortic disease | Yes | 23 (4.3%) |
| | No | 87 (16.2%) |
| | No information | 428 (79.6%) |
| History of smoking | Yes | 197 (36.6%) |
| | No | 104 (19.3%) |
| | No information | 237 (44.1%) |

AAD, acute aortic dissection

**Table 2. Association between meteorological factors and AAD onset.**

| | Case days | Control days | | |
|---|---|---|---|---|
| | Mean (± SD) | Mean (± SD) | OR (95% CI) | P-value |
| Daily mean relative humidity (%) | 65.77 ± 15.33 | 66.70 ± 15.10 | 1.01 (1.001.02) | 0.205 |
| Daily mean wind speed (m/s) | 2.15 ± 0.72 | 2.19 ± 0.76 | 1.08 (0.921.27) | 0.347 |
| Sunshine duration (h) | 6.36 ± 4.00 | 6.05 ± 4.05 | 0.98 (0.951.01) | 0.164 |
| Daily mean station pressure (hPa) | 1012.33 ± 6.22 | 1012.74 ± 6.09 | 1.01 (0.991.04) | 0.213 |
| Daily mean temperature (˚C) | 15.43 ± 7.45 | 15.88 ± 7.23 | 1.10 (1.041.16) | 0.001 |
| Daily minimum temperature (˚C) | 11.36 ± 8.06 | 11.85 ± 7.95 | 1.09 (1.031.14) | 0.001 |
| Daily maximum temperature (˚C) | 19.96 ± 7.26 | 20.30 ± 6.96 | 1.05 (1.001.10) | 0.038 |
| Diurnal temperature range (˚C) | 8.60 ± 2.92 | 8.45 ± 2.94 | 0.98 (0.931.02) | 0.349 |
| Difference in the daily mean temperature from the previous day (˚C) | 0.15 ± 1.88 | -0.09 ± 1.98 | 0.96 (0.901.02) | 0.165 |
| Mean of the daily mean temperatures for the previous 3 days (˚C) | 15.57 ± 7.25 | 15.86 ± 7.11 | 1.10 (1.031.18) | 0.007 |
| Mean of the daily mean temperatures for the previous 5 days (˚C) | 15.57 ± 7.15 | 15.88 ± 7.06 | 1.14 (1.051.24) | 0.001 |
| Mean of the daily mean temperatures for the previous 7 days (˚C) | 15.55 ± 7.14 | 15.85 ± 7.05 | 1.17 (1.071.28) | <0.001 |

AAD, acute aortic dissection; SD, Standard Deviation; OR, Odds Ratio; CI, Confidence Interval

The significance level was set at 0.050.

**Table 3. Association between the mean of the daily mean temperatures for the previous 7 days and AAD onset by subset analysis.**

| | | Case days | Control days | | |
|---|---|---|---|---|---|
| | n (%) | Mean (± SD) | Mean (± SD) | OR (95% CI) | P-value |
| All | 538 (100%) | 15.43 ± 7.45 | 15.88 ± 7.23 | 1.17 (1.071.28) | <0.001 |
| Men | 307 (57%) | 15.55 ± 7.29 | 15.97 ± 7.31 | 1.25 (1.111.42) | <0.001 |
| Women | 231 (43%) | 15.28 ± 7.66 | 15.76 ± 7.14 | 1.08 (0.941.23) | 0.267 |
| High temperature period (June–September) | 130 (24%) | 25.46 ± 2.86 | 25.52 ± 2.82 | 0.99 (0.801.23) | 0.936 |
| Low temperature period (December–March) | 221 (41%) | 8.50 ± 3.44 | 9.14 ± 3.20 | 1.27 (1.101.46) | 0.001 |
| Under 65 years old | 170 (32%) | 16.01 ± 7.91 | 16.37 ± 7.64 | 1.10 (0.941.28) | 0.242 |
| 65–74 years old | 147 (27%) | 14.07 ± 7.17 | 14.83 ± 6.89 | 1.29 (1.081.54) | 0.006 |
| Over 75 years old | 221 (41%) | 15.89 ± 7.17 | 16.20 ± 7.09 | 1.17 (1.021.34) | 0.028 |
| Type A dissection | 334 (62%) | 15.29 ± 7.42 | 15.65 ± 7.12 | 1.08 (0.961.21) | 0.190 |
| Type B dissection | 195 (36%) | 15.67 ± 7.52 | 16.27 ± 7.44 | 1.33 (1.141.54) | <0.001 |
| History of smoking | 197 (37%) | 16.27 ± 7.51 | 16.59 ± 7.40 | 1.31 (1.121.53) | 0.001 |
| Complications of hypertension | 158 (29%) | 15.27 ± 7.29 | 15.82 ± 6.92 | 1.25 (1.061.48) | 0.010 |
| Family history of aortic disease | 23 (4%) | 15.31 ± 8.50 | 15.60 ± 7.66 | 1.12 (0.761.65) | 0.564 |
| Time of AAD onset of 0–6 am | 67 (12%) | 16.87 ± 8.26 | 17.74 ± 7.56 | 1.29 (0.981.69) | 0.069 |
| Time of AAD onset of 6 am–12 pm | 135 (25%) | 14.14 ± 7.06 | 14.70 ± 6.96 | 1.24 (1.041.47) | 0.015 |
| Time of AAD onset of 12 pm–6 pm | 118 (22%) | 15.09 ± 7.55 | 15.45 ± 7.07 | 1.10 (0.931.31) | 0.273 |
| Time of AAD onset of 6 pm–12 am | 92 (17%) | 15.51 ± 7.40 | 15.70 ± 7.12 | 1.00 (0.781.30) | 0.975 |
| Men with a history of smoking | 169 (31%) | 15.99 ± 7.32 | 16.43 ± 7.36 | 1.35 (1.151.59) | <0.001 |
| Men with hypertension | 77 (14%) | 16.26 ± 7.23 | 16.91 ± 6.81 | 1.40 (1.091.81) | 0.010 |
| Aged 65–74 years with hypertension | 42 (8%) | 14.63 ± 7.06 | 15.54 ± 6.58 | 1.50 (1.012.24) | 0.047 |

AAD, acute aortic dissection; SD, Standard Deviation; OR, Odds Ratio; CI, Confidence Interval

The significance level was set at 0.050.

"men with hypertension" (OR = 1.40; 95% CI = 1.09–1.81), and "aged 65–74 years with hypertension" (OR = 1.50; 95% CI = 1.01–2.24).

## Discussion

In this study, we found that the daily mean temperature on the day of AAD onset was lower than that on control days. Furthermore, the risk of AAD onset increased when the daily mean temperature remained low for 7 days. In the subset analysis by patient background, the risk of AAD onset increased by 1.35 times in male smokers, 1.40 times in men with hypertension, and 1.50 times in adults aged 65–74 years with hypertension when there was a 1˚C decrease in the daily mean temperature.

In this study, the highest onset of AAD was during winter. The risk of AAD onset increased during low temperature periods (OR = 1.27). Previous studies have also reported that AAD onset was more frequent in low temperature periods [4, 8, 9]. One reason for the high incidence of AAD onset during the winter is exposure to cold temperatures. Exposure to cold temperatures can cause vascular damage in the aorta due to the influence of sympathetic nervous activity or increased blood pressure, which strengthens the forces that deform the vascular wall, including friction inside the vessels [18]. Such biological reactions can trigger the onset of AAD.

The time of AAD onset was most frequent between 6:00 am and 12:00 pm and least frequent between midnight and 6:00 am. A previous study also found that AAD onset was most common between 6:00 am and 12:00 pm [8]. At this time, people generally wake up and begin physical activity. Morning blood pressure surge, which is an increase in blood pressure in the morning, may have an effect. Morning blood pressure is thought to be a phenomenon in which blood pressure increases in the morning due to increased sympathetic nervous system activity around the time of awakening and changes in the autonomic nervous system activity [18, 19]. In addition, patients receiving oral treatment for hypertension may also be affected by the tendency of blood pressure to rise in the morning due to the reduced effect of antihypertensive medications.

In this study, we analyzed the association between the mean of the daily mean temperature of the previous 3, 5, and 7 days and AAD onset. The AAD onset was most strongly associated with the mean of the daily mean temperature of the previous 7 days. The risk of AAD onset increased by 1.17 times when the mean of the daily mean temperature over 7 days decreased by 1˚C. This study's results suggest that significantly different temperatures and sustained low-temperature days are associated with AAD onset. There was no observed association between meteorological factors other than temperature and AAD onset, including daily mean relative humidity, daily mean wind speed, sunshine duration, and daily mean station pressure. A study conducted in Aichi Prefecture, Japan, reported that temperature was associated with the onset of AAD, which is similar to the results of this study [8]. Aichi Prefecture is adjacent to Shizuoka Prefecture, where this study was conducted, and has similar weather conditions; thus, the results being consistent with the results of the present study help validate the results of the present study.

Information about individuals at high risk of onset of AAD in certain weather conditions can be useful for the reduction and awareness of the onset of AAD. In Japan, AAD is more frequent in men, and the peak age of onset of AAD in men and women is in their 70s. Further, hypertension and smoking [1, 20, 21] are risk factors for the onset of AAD. A subset analysis by patient background showed that the risk of AAD onset increased when the mean of the daily mean temperature over 7 days decreased by 1˚C in men, Stanford B type, those with a history of smoking, and those with hypertension. The results for men, those with a history of smoking, and those with hypertension were consistent with previous studies in Japan [1]. No

previous study has reported an association between the mean of the daily mean temperature over 7 days decreased by 1°C and Stanford Type B; however, the effects of meteorological factors may differ depending on the type of AAD. The following reasons may account for the significantly increased ORs observed in older participants aged 65 to 74 in this study. Older people aged 65 to 74 are more likely to be exposed to cold because they may be more likely to work or be active compared to adults older than them. Therefore, they may be more susceptible to the effects of meteorological factors. In contrast, patients under 65 are not significantly affected by meteorological factors but may be affected by other factors, such as genetic factors. In addition, older patients over 75 may not have an increased risk of AAD onset because of reduced activity and less exposure to cold, owing to decreased physical capacity.

In this study, we excluded patients who developed AAD in the hospital because of the unique circumstances of being under medical care. Of the seven excluded cases, five developed AAD during surgery. This study employed a self-controlled design, where the case was the patient at the time of AAD onset, and the control was the same patient at a randomly selected time within a 29-day period centered around the AAD onset. This means that all variables were matched except for the presence or absence of AAD and meteorological factors. Consequently, multivariate analysis could not be performed, and the analysis was conducted by repeating stratified univariate analyses. In future research, multivariate analysis will be possible by using appropriate controls (other than self-controls) for AAD cases. If multivariate analysis can be performed, it will allow for the simultaneous examination of multiple factors.

This study has some limitations. First, the study did not include patients who died before arriving at the hospital, potentially leading to an underrepresentation of individuals who developed AAD but were not identified in this study. Second, incomplete information was obtained from the electronic medical records in some cases. As shown in Table 1, missing values were present in 79.6% of cases for family history, 60.4% for complications, and 44.1% for smoking history. This may be due to the emergency nature of AAD, which leads to emergency interventions, such as cardiopulmonary resuscitation and surgery, upon hospital arrival. The necessity of emergency care may result in skipping the collection of certain background factors. To gather information that was not recorded in the medical records, it would have been necessary to collect data directly from patients and their families, which was not feasible because of budget constraints. Therefore, the results of the subset analysis using these factors may contain bias and should be interpreted with caution. Lastly, information on housing and clothing at the time of AAD onset was unavailable from the electronic medical records, thus preventing the evaluation of these factors.

As future perspectives of this study, it is necessary to verify whether the associations between meteorological factors and AAD onset observed in this research can be observed in other regions or countries. Moreover, by obtaining data that could not be captured from electronic medical records through alternative methods, it is possible to make the evidence of the association between meteorological factors and AAD onset more robust. We believe that we can alert people to take precautions to prevent the onset of AAD during periods when temperatures are expected to remain cool. This may be especially beneficial for those at high risk of AAD.

## Conclusion

This study found that the risk of AAD onset increased with sustained low-temperature days. In particular, when the daily mean temperature remained low for a week, the risk of AAD onset increased in men with a history of smoking and hypertension and older people aged 65–74 with hypertension. The results of this study are expected to help raise awareness in clinical

practice and among the general public to prevent the onset of cardiovascular diseases, such as myocardial infarction, heart failure, and AAD, which are influenced by a drop in temperature.

## Acknowledgments

We are deeply grateful to the patients who provided valuable data for this study, everyone at Shizuoka City Shizuoka Hospital, and those involved in this research. We also thank Editage (www.editage.jp) for their English language editing services.

## Author Contributions

**Conceptualization:** Ayami Ishikawa, Yasuto Sato, Takeshi Usui.

**Data curation:** Ayami Ishikawa.

**Formal analysis:** Ayami Ishikawa.

**Methodology:** Yasuhiko Terai.

**Project administration:** Yasuto Sato.

**Supervision:** Yasuhiko Terai, Takeshi Usui.

**Writing – original draft:** Ayami Ishikawa.

**Writing – review & editing:** Yasuto Sato.

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
