## [Decision Letter · Decision Letter 0]

6 Aug 2024

PONE-D-24-23649Epidemiological study of the relationship between meteorological factors and onset of acute aortic dissection in JapanPLOS ONE

Dear Dr. Sato,

Thank you for submitting your manuscript to PLOS ONE. After careful consideration, we feel that it has merit but does not fully meet PLOS ONE’s publication criteria as it currently stands. Therefore, we invite you to submit a revised version of the manuscript that addresses the points raised during the review process.

We look forward to receiving your revised manuscript.

Kind regards,

Chiara Lazzeri

Academic Editor

PLOS ONE

Journal Requirements:

Reviewers' comments:

Reviewer's Responses to Questions

**Comments to the Author**

1. Is the manuscript technically sound, and do the data support the conclusions?

Reviewer #1: Yes

Reviewer #2: Yes

2. Has the statistical analysis been performed appropriately and rigorously? 

Reviewer #1: Yes

Reviewer #2: Yes

3. Have the authors made all data underlying the findings in their manuscript fully available?

Reviewer #1: No

Reviewer #2: Yes

4. Is the manuscript presented in an intelligible fashion and written in standard English?

Reviewer #1: Yes

Reviewer #2: Yes

5. Review Comments to the Author

Reviewer #1: “Epidemiological study of the relationship between meteorological factors and onset of acute aortic dissection in Japan”

This article is aimed to deepen the knowledge of meteorological data and its changes and their correlation with aortic dissection incidence. As weather differentiates worldwide the authors correctly focused their analysis on a specific region’s weather data (Shizuoka Prefecture, Japan), providing an overview of possible external factors leading to AAD in their climate zone.

No significant similarities with other articles published can be found, with a similarity index close to 3%.

The article is well written. No major grammatical errors have been identified.

The referral population is correctly identified. Subdivision on several groups based on preoperative, anamnestic and clinical factors has been correctly made. Uncertainties remains upon data collection on patient’s medical history. More information on family history for aortic disease and patient’s smoking history could be obtained to emphasize some of the results that authors identified, such as correlation between meteorological changes and incidence of AAD in smoker patients.

Statistical analysis is correctly achieved. Logistic regression has been used to analyze the correlation between meteorological changes and AAD incidence in the various groups, permitting to identify variables correlated with an increased incidence of aortic dissection with statistical significance. As suggestion, multivariate analysis should be implemented to identify correlation between variables identified to be the major responsible for aortic dissection onset.

I may also suggest highlighting p-value significance values at the bottom of tables 2 and 3 (p-value < 0.05 for CI of 95%).

Authors’ conclusions are sustained by what has been evaluated in the statistical analysis, but further analysis should be obtained as suggested before.

The article permits an overview and significant statistical data upon a well-known topic that many authors already discussed before, leading to a deeper perception of a high-risk pathology and its triggers. For this it should be accepted after a minor/major review with implementation of what has been suggested.

Reviewer #2: In this manuscript, the authors aimed to identify meteorological factors associated with the onset of acute aortic dissection (AAD). In their single-center, retrospective analysis of 538 patients, they note that AAD onset was associated with a lower daily mean temperature in the preceding 7 days. This effect was compounded by traditional risk factors (i.e., hypertension, smoking, male sex).

Major Comments

1. How the time of onset of AAD was defined requires clarification. Was time of AAD onset defined as the time of onset of chest pain as reported by the patient upon hospital admission? Or was time of AAD onset defined as the time of hospital admission? If the latter is the case, then a clear gap is present. I would then suggest revising the definition to the former as this would more accurately represent time of AAD onset.

2. Although it is mentioned by the authors in the limitations section of the manuscript, I do think it is prudent to highlight the lack of information available for select baseline characteristics (medications, complications, family history, smoking, etc.). I understand that this is an inherent limitation of the retrospective study design and criticality of the condition; however, when information is available for ~50% or less of the patients then one must question the validity of select sub-analyses that were performed.

Minor Comments

1. I would suggest revising the final statement in the abstract to remained focused on AAD rather than various cardiovascular disease states.

2. The last statement in the introduction should simply state the objective of the manuscript. The authors should avoid stating any results/conclusions of the study at this point.

3. Why were patients with AAD onset in the hospital omitted? Does this refer to patients with iatrogenic aortic dissections? If not, then it is unclear why these patients would be omitted. I presume it is secondary to a lack of outdoor exposure in hospitalized patients; however, the assumption is then made that all patients who suffer AAD outside of hospital are having similar exposure.

4. The authors should define “complications” in the data extraction section of the manuscript. Furthermore, they should better define malperfusion – cerebral, coronary, visceral, spinal cord, upper/lower extremity, etc.

5. References should be provided to support the following statement – “Meteorological factors influencing the onset of AAD include seasonal, weekly, and intraday fluctuations in temperature, humidity, atmospheric pressure, and wind speed.”

6. PLOS authors have the option to publish the peer review history of their article (what does this mean?). If published, this will include your full peer review and any attached files.

Reviewer #1: **Yes: **Prof. Pierluigi Stèfano

Reviewer #2: No

---

## [Author Response · Author response to Decision Letter 0]

17 Sep 2024

Thank you very much for reviewing our submission. We have revised our manuscript in accordance with the reviewers’ suggestions. Our point-by-point responses are highlighted in yellow below.

Journal Requirements:

Thank you very much for reviewing our manuscript.

We have added the following to the Ethical considerations section and the online submission information:

The ethical committee approved the use of an opt-out consent process for study participation. An opt-out document was posted on the hospital's website, and individuals who did not agree to participate were excluded from the study.

Thank you for your valuable comments. We have added the following to the online submission information:

As the dataset for this study contains patient personal information, there are restrictions on sharing the data. If you wish to obtain the dataset, you will need to submit a reasonable request to the corresponding author. Upon receiving this request, the ethical review committee will conduct an individual review.

Reviewers' comments:

Reviewer's Responses to Questions

Comments to the Author

1. Is the manuscript technically sound, and do the data support the conclusions?

Reviewer #1: Yes

Reviewer #2: Yes

Thank you for taking the time to review our submission.

2. Has the statistical analysis been performed appropriately and rigorously?

Reviewer #1: Yes

Reviewer #2: Yes

3. Have the authors made all data underlying the findings in their manuscript fully available?

Reviewer #1: No

Reviewer #2: Yes

Thank you for your perceptive comments. As mentioned above, we have outlined the procedure for obtaining the dataset.

4. Is the manuscript presented in an intelligible fashion and written in standard English?

Reviewer #1: Yes

Reviewer #2: Yes

5. Review Comments to the Author

Reviewer #1: “Epidemiological study of the relationship between meteorological factors and onset of acute aortic dissection in Japan”

This article is aimed to deepen the knowledge of meteorological data and its changes and their correlation with aortic dissection incidence. As weather differentiates worldwide the authors correctly focused their analysis on a specific region’s weather data (Shizuoka Prefecture, Japan), providing an overview of possible external factors leading to AAD in their climate zone.

No significant similarities with other articles published can be found, with a similarity index close to 3%.

The article is well written. No major grammatical errors have been identified.

The referral population is correctly identified. Subdivision on several groups based on preoperative, anamnestic and clinical factors has been correctly made. Uncertainties remains upon data collection on patient’s medical history. More information on family history for aortic disease and patient’s smoking history could be obtained to emphasize some of the results that authors identified, such as correlation between meteorological changes and incidence of AAD in smoker patients.

Thank you for reviewing our manuscript.

To gather information not recorded in the medical records, it would have been necessary to collect data directly from patients and their families, which was not feasible because of budget constraints.

We have added a note about this in the Discussion section.

Statistical analysis is correctly achieved. Logistic regression has been used to analyze the correlation between meteorological changes and AAD incidence in the various groups, permitting to identify variables correlated with an increased incidence of aortic dissection with statistical significance. As suggestion, multivariate analysis should be implemented to identify correlation between variables identified to be the major responsible for aortic dissection onset.

Thank you for your insightful comments. This study used a self-controlled design, where the case was the patient at the time of AAD onset, and the control was the same patient at a randomly selected time within a 29-day period centered around the AAD onset. This means that all variables were matched except for the presence or absence of AAD and meteorological factors. Consequently, multivariate analysis could not be performed, and analysis was conducted using repeating stratified univariate analysis. In future research, multivariate analysis will be possible by using appropriate controls (other than self-controls) for AAD cases. This will allow for the simultaneous examination of multiple factors.

We have added a note about this in the Discussion section.

I may also suggest highlighting p-value significance values at the bottom of tables 2 and 3 (p-value < 0.05 for CI of 95%).

Thank you for pointing this out. Significance levels are provided in Tables 2 and 3.

Authors’ conclusions are sustained by what has been evaluated in the statistical analysis, but further analysis should be obtained as suggested before.

The article permits an overview and significant statistical data upon a well-known topic that many authors already discussed before, leading to a deeper perception of a high-risk pathology and its triggers. For this it should be accepted after a minor/major review with implementation of what has been suggested.

Reviewer #2: In this manuscript, the authors aimed to identify meteorological factors associated with the onset of acute aortic dissection (AAD). In their single-center, retrospective analysis of 538 patients, they note that AAD onset was associated with a lower daily mean temperature in the preceding 7 days. This effect was compounded by traditional risk factors (i.e., hypertension, smoking, male sex).

Major Comments

1. How the time of onset of AAD was defined requires clarification. Was time of AAD onset defined as the time of onset of chest pain as reported by the patient upon hospital admission? Or was time of AAD onset defined as the time of hospital admission? If the latter is the case, then a clear gap is present. I would then suggest revising the definition to the former as this would more accurately represent time of AAD onset.

Thank you for reviewing our manuscript.

The time of AAD onset was determined based on the "patient-reported symptom onset time" as documented in the initial medical record.

We have added this note to the Data collection section.

2. Although it is mentioned by the authors in the limitations section of the manuscript, I do think it is prudent to highlight the lack of information available for select baseline characteristics (medications, complications, family history, smoking, etc.). I understand that this is an inherent limitation of the retrospective study design and criticality of the condition; however, when information is available for ~50% or less of the patients then one must question the validity of select sub-analyses that were performed.

We appreciate your meaningful comments. As shown in Table 1, missing values occurred in 79.6% of cases for family history, 60.4% for complications, and 44.1% for smoking history.

Therefore, the results of the subset analysis using these factors may contain bias and should be interpreted with caution.

We have added this note to the Discussion section.

Minor Comments

1. I would suggest revising the final statement in the abstract to remained focused on AAD rather than various cardiovascular disease states.

Thank you for bringing this to our attention. The text has been revised.

The results of this study are expected to help raise awareness in clinical practice and among the general public about reducing the onset of AAD, which is influenced by a drop in temperature.

2. The last statement in the introduction should simply state the objective of the manuscript. The authors should avoid stating any results/conclusions of the study at this point.

We appreciate your attention to this matter. The text has been revised.

This study aimed to identify meteorological factors associated with the onset of AAD in Japan.

3. Why were patients with AAD onset in the hospital omitted? Does this refer to patients with iatrogenic aortic dissections? If not, then it is unclear why these patients would be omitted. I presume it is secondary to a lack of outdoor exposure in hospitalized patients; however, the assumption is then made that all patients who suffer AAD outside of hospital are having similar exposure.

Thank you for your important comments. In this study, we excluded patients who developed AAD in the hospital because of the unique circumstances of being under medical care. Of the seven excluded cases, five developed AAD during surgery.

We have added this note to the Discussion section.

4. The authors should define “complications” in the data extraction section of the manuscript. Furthermore, they should better define malperfusion – cerebral, coronary, visceral, spinal cord, upper/lower extremity, etc.

Your meaningful comment is duly noted. For complications, data were collected on cardiopulmonary arrest, aortic rupture, cardiac tamponade, and malperfusion. Cardiopulmonary arrest was considered present if it was noted at the time of transport to the hospital. Aortic rupture, cardiac tamponade, and malperfusion were considered present if they were noted in the medical record at the time of admission. For malperfusion, only the presence or absence was reported, and the location could not be specified.

We have added this note to the Methods section.

5. References should be provided to support the following statement – “Meteorological factors influencing the onset of AAD include seasonal, weekly, and intraday fluctuations in temperature, humidity, atmospheric pressure, and wind speed.”

Your insightful comment is appreciated. We have added the references to the text.

---

## [Editor Report · Decision Letter 1]

20 Sep 2024

Epidemiological study of the relationship between meteorological factors and onset of acute aortic dissection in Japan

PONE-D-24-23649R1

Dear Dr. Sato,

We’re pleased to inform you that your manuscript has been judged scientifically suitable for publication and will be formally accepted for publication once it meets all outstanding technical requirements.

Kind regards,

Chiara Lazzeri

Academic Editor

PLOS ONE
---

## [Editor Report · Acceptance letter]

30 Sep 2024

PONE-D-24-23649R1 

PLOS ONE

Dear Dr. Sato, 

I'm pleased to inform you that your manuscript has been deemed suitable for publication in PLOS ONE. Congratulations! Your manuscript is now being handed over to our production team.

Kind regards, 

on behalf of

Dr. Chiara Lazzeri 

Academic Editor

PLOS ONE